# The Effects of the Use of Plyometric Exercises with and without the Ball in the Development of Explosive Strength in Volleyball

**DOI:** 10.3390/jfmk9030126

**Published:** 2024-07-18

**Authors:** Giovanni Esposito, Gaetano Altavilla, Giuseppe Giardullo, Rosario Ceruso, Tiziana D’Isanto

**Affiliations:** 1Research Centre of Physical Education and Exercise, Pegaso University, 80143 Napoli, Italy; gioesposito@unisa.it (G.E.); gaetano.altavilla@unipegaso.it (G.A.); giardu@hotmail.com (G.G.); tizidisanto@gmail.com (T.D.); 2Department of Human, Philosophical and Education, University of Salerno, 84084 Fisciano, Italy; 3Department of Neuroscience, Biomedicine and Movement, University of Verona, 37129 Verona, Italy

**Keywords:** explosive strength, jump, types of training, performance

## Abstract

Volleyball primarily focuses on technical and tactical training with a ball. However, there is growing interest in integrating fitness training into volleyball practice, particularly to enhance explosive strength through plyometric methods, but there is a lack of a direct scientific comparison between training with and without the ball. This study aimed to compare the effects of two training protocols on volleyball players. One protocol combined plyometric exercises with technical gestures (wall drills) using the ball, while the other protocol excluded the ball during plyometric exercises. Twenty male volleyball players (aged 18.6 ± 0.3 years, height 189.8 ± 2.2 cm, weight 79.4 ± 1.6 kg) were divided into experimental (with ball) and control (without ball) groups. The analysis of the results highlights significant improvements in both the squat jumps (SJs) and the countermovement jumps with arm swing (CMJas) for both groups. While there were no significant differences between the groups for SJ, significant differences emerged in CMJas, indicating varied training effects. Specifically, the interaction effect was significant (*p* = 0.004), demonstrating a meaningful distinction in performance improvements between the two groups. The effect size of the interaction is moderate (η_p_^2^ = 0.37, 95% CI: 0; 0.91). These results suggest that incorporating a ball into plyometric training can be beneficial for developing explosive strength in a different way, thereby improving performance due to the motivational stimulus provided. However, given the specificity of the sample and the training protocols used, further studies are needed to confirm these results and evaluate their applicability to a larger sample of volleyball players.

## 1. Introduction

Volleyball is an intermittent and intense anaerobic team sport that combines explosive movements in both vertical and horizontal directions with short periods of recovery [1]. As a dynamic and fast-paced sport, volleyball requires athletes to constantly adapt to rapidly changing situations on the court. This adaptability classifies volleyball as an open-skill sport, where players must perform a variety of acyclic movements [2]. These movements demand a high level of technical proficiency and physical conditioning, making volleyball a sport where both skill and athleticism are paramount. Players must develop excellent sport-specific skills, including precision in passing, setting, spiking, and blocking. Additionally, they must possess quick reaction abilities to respond to different technical and tactical stimuli in very short time frames [3]. The combination of these requirements means that volleyball players need to be well-rounded athletes with superior physical and cognitive skills [4]. One of the most defining aspects of a volleyball player’s performance is their jumping ability, which is essential for both offensive and defensive actions such as spiking and blocking [5]. Modern high-level competitive volleyball has seen an increased emphasis on the use of the jump, reflecting a clear trend towards faster gameplay and higher physical demands [6,7,8]. This shift necessitates the development of strength in its various forms, particularly explosive strength, which is critical for achieving maximum height and power during jumps [9].

To effectively meet these demands, volleyball players need to improve several fundamental physiological determinants, including different types of strength. Explosive strength, rapid strength, and reactive strength are particularly important in volleyball, as they underpin many of the sport’s key actions [10]. Targeted training protocols, both with and without the ball, are employed to enhance these attributes and improve overall performance [11]. Plyometric training has been identified as a critical component in the development of these strength qualities. This type of training focuses on exercises that enhance the explosive power of muscles through rapid and powerful movements utilizing the stretch–shortening cycle [12,13]. Plyometric exercises are highly effective for fast strength athletes, as they simulate the quick, powerful actions required in sports like volleyball [14]. In addition to improving performance, enhancing explosive strength through plyometric training plays a significant role in injury prevention. By strengthening muscles and improving their responsiveness, athletes can reduce the risk of common injuries associated with the high-impact nature of volleyball [15].

Despite the recognized benefits of plyometric training in enhancing explosive strength and performance in volleyball [16], there remains a gap in understanding the differential effects of incorporating the ball into these training protocols. Previous studies have established the general efficacy of plyometric exercises but have not specifically addressed whether the inclusion of the ball during training confers additional benefits in terms of performance and injury prevention [17,18,19]. Recent studies have shown that jump-specific plyometric training, integrated with technical elements of the game, can lead to significant improvements in jump performance among volleyball players [20,21]. A study conducted by Marques et al. [22] confirmed the benefits of plyometric training on explosive strength and jump height in volleyball players. Similarly, research by Sheppard et al. [23] has demonstrated that plyometric training can significantly enhance jump power in volleyball players. Saez-Saez de Villarreal et al. [24] evaluated the effectiveness of plyometric protocols and found improvements in explosive strength and speed.

This gap in the literature necessitates further investigation to optimize training programs for volleyball players. Therefore, the primary objective of this study is to investigate the effects of plyometric training on volleyball players, particularly focusing on exercises that incorporate technical gestures such as blocking. The study compares two different training protocols: one that includes the use of the ball during plyometric exercises and one that does not. By examining these protocols, the study aims to determine the most effective methods for enhancing the performance of volleyball players. We hypothesize that the use of the ball in plyometric training leads to an increase in jump heights, directly caused by the addition of the ball in plyometric training.

## 2. Materials and Methods

### 2.1. Study Participants

The sample consisted of 20 young male volleyball players (positions: Central, Opposite, and Spiker) aged 18.6 ± 0.3 years, with a height of 189.8 ± 2.2 cm, weight of 79.4 ± 1.6 kg, and BMI of 22.0 ± 1.2. These players participated in the Italian Volleyball Championship (Under 19 Category). All players participated voluntarily and were informed about the experimental protocol and the study’s objectives. They were familiar with the procedures used. The inclusion criterion was a minimum of five years of sports experience, and the exclusion criterion was any muscle, tendon, or bone injuries reported in the previous 12 months.

### 2.2. Procedure

Participants, all from the same volleyball team, were assigned to two groups of 10 players each through simple randomization: an experimental group (with the ball) and a control group (without the ball). The participant characterization began with the use of the Pegaso digital scale (GIMA Spa, Gessate, Italy) to gather accurate anthropometric data, including height and weight, which formed the foundational dataset for subsequent analyses. A plyometric training protocol was applied before the start of the championship.

The experimental group underwent a specialized plyometric training program designed to integrate the technical execution of blocking (wall) with ball handling skills. This group engaged in three training sessions per week over a period of 6 weeks. Each session included a dynamic warm-up, core stability exercises, and a series of jumps aimed at enhancing explosive power and coordination. During the initial 3-week phase, participants performed 3 sets of 8 repetitions of jumps incorporating the wall technique with a volleyball. In the subsequent 3 weeks, this was intensified to 4 sets of 8 repetitions per session. The training began with a dynamic warm-up lasting approximately 15 min, including exercises such as high knees, butt kicks, and dynamic stretches to prepare the muscles for high-intensity activity [25]. The core stability exercises included planks, Russian twists, and leg raises, aimed at improving the players’ overall stability and control [26]. The key component of the experimental group’s protocol involved performing the wall technique with a volleyball. Players executed a jump followed by a quick bounce to block a ball thrown by a coach’s assistant positioned above the net at approximately 3 m. The assistant consistently maintained a throwing height of 50 cm above the net and ensured the ball’s trajectory aligned with the player’s ascending phase of the jump. This method was carefully designed to simulate game-like conditions, enhancing players’ ability to perform explosive movements and react quickly to the ball [27]. The goal was to optimize take-off speed and technical proficiency in blocking.

The control group followed an identical plyometric training program to the experimental group, but without the integration of ball handling. Like the experimental group, they participated in three weekly sessions spanning 6 weeks, each involving a dynamic warm-up, core stability exercises, and a series of jumps designed to enhance explosive strength and agility. During the training sessions, participants performed the same number of sets and repetitions as the experimental group, focusing solely on plyometric exercises without incorporating the ball. This standardized approach ensured that both groups experienced similar training intensities and conditions, allowing for a direct comparison of the intervention’s impact on performance metrics [28]. The graphic representation of the training protocol is shown in Table 1.

### 2.3. Data Collection and Measurement

Throughout the study period, data were collected using standardized methods for vertical jump performance, specifically the squat jump (SJ) and countermovement jump with arm swing (CMJas). These measurements were conducted using the Optojump System (Microgate Srl, Bolzano, Italy) to accurately capture changes in jump height and technique [29]. This system, comprising a transmitter and receiver bar positioned one meter apart, utilized optical sensing technology to measure jump height with high resolution (1.041 cm).

The SJ and CMJas tests were performed according to the procedures outlined by Bosco et al. [30]. As in previous studies [31,32,33], the study was conducted in a controlled laboratory environment measuring 6 × 4 × 4 m (length × width × height). Specialized PVC flooring was chosen to minimize external variables affecting jump performance. An acquisition area of 90 × 60 cm (length × width), housing the Optojump system, was marked on the laboratory floor to confine participants’ movements and enhance test reproducibility. Pre-training and post-training assessments were conducted to evaluate improvements in jump performance metrics across both groups. This comprehensive evaluation included measuring jump height, ground contact time, and reactive strength index, providing a thorough analysis of the intervention’s effectiveness. To ensure consistency and reliability of data, all tests were administered by the same trained personnel, and players were given standardized instructions and encouragement during the tests. Additionally, players were familiarized with the testing procedures before the initial assessments to minimize learning effects. Each participant performed three trials of each jump test with a 120 s rest period between trials to ensure robust dataset collection. The average result of these trials was recorded for subsequent statistical analysis.

### 2.4. Statistical Analysis

The data reported in descriptive form are expressed with mean and standard deviation values (mean ± SD). The normality of the data distribution was verified using the Shapiro–Wilk test, while the homogeneity of the variances was verified with the Levene test. An independent-sample *t*-test was performed to assume non-significant differences between values before the training program. A two-way repeated-measures analysis of variance was used to test for differences in training induced changes in performance variables. The independent variables included one between-subjects factor (training intervention) with two levels (Experimental and Control group) and one within-subjects factor (time) with two levels (pre-and post-intervention). To qualitatively interpret the magnitude of differences, effect sizes and associated 95% CI were classified as small (0.2–0.5), moderate (0.5–0.8), and large (>0.8) [34]. The significance level was set at *p* < 0.05. All statistical tests were processed using IBM SPSS (version 22; IBM, Armonk, NY, USA).

## 3. Results

The anthropometric characteristics of the participants are summarized in Table 2.

Variables highlighting homogeneous average values. This ensures that any observed differences in performance can be attributed to the experimental conditions rather than variations in anthropometric characteristics. A well-matched sample is critical for minimizing confounding factors in performance outcomes.

The initial performance metrics for squat jump (SJ) and countermovement jump with arm swing (CMJas) are presented in Table 3.

There were no significant differences between the groups at the start of the study, indicating that the groups were comparable at baseline. This comparability ensures that any subsequent changes in performance can be attributed to the training interventions rather than pre-existing differences.

Table 4 shows the performance metrics for SJ and CMJas before and after the training period.

The analysis of vertical jump performance results at the end of two plyometric training protocols—one involving the use of the ball (experimental group) and the other without the ball (control group)—revealed significant improvements in both groups for SJ and CMJas. Specifically, the improvements in SJ performance in the experimental group were significant (*p* = 0.001) with an effect size of d = 0.98 (95% CI: 0.72; 0.99). Similarly, the control group showed significant improvements (*p* = 0.001) with an effect size of d = 0.99 (95% CI: 0.94; 0.99).

No significant differences were found between the experimental group and the control group regarding SJ improvements measured at different times. In essence, both groups demonstrated similar improvements in this type of jump. The small effect size (η_p_^2^ = 0.09) suggests that plyometric training, with or without a ball, had a comparable impact on jumping performance for both groups. Furthermore, the confidence intervals (95% CI: 0; 0.76) indicate some variability and uncertainty in the measurement of the effect.

Regarding the CMJas, the experimental group showed significant improvements (*p* = 0.001) with an effect size of d = 0.98 (95% CI: 0.76; 0.99), whereas the control group showed similarly significant improvements (*p* = 0.001) with an effect size of d = 0.97 (95% CI: 0.63; 0.99). However, in this case, the interaction between measurement time and group was significant (*p* = 0.004), indicating that the changes in performance in countermovement jumping with arm swing were different between the two groups. The effect size of the interaction was moderate (η_p_^2^ = 0.37), although it was associated with significant uncertainty (95% CI: 0; 0.91).

## 4. Discussion

This study compares two distinct plyometric training protocols aimed at enhancing performance among volleyball players: one that integrates the use of a ball during exercises, and another that excludes it. The initial data analysis did not reveal significant differences among participants, enabling the formation of well-matched experimental and control groups. During the structured six-week intervention period, both groups demonstrated significant improvements in jumping ability (SJ and CMJas). Specifically, the experimental group utilizing the ball showed improvements of 2.4% in SJ and 2.2% in CMJas, while the control group exhibited slightly smaller improvements of 2.2% and 1.9%, respectively. There were no significant differences between the groups for the SJ, indicating a similar effect of the training. However, for the CMJas, both groups exhibited significant improvements, though with variations in the results, as indicated by a significant interaction (*p* = 0.004).

The results obtained support the existing literature on the effectiveness of plyometric training in enhancing muscle power and explosiveness, which are crucial for volleyball movements. Previous studies, such as those by Markovic and Mikulic [35], have documented significant improvements in jumping ability following plyometric interventions, emphasizing the importance of this type of training for athletes involved in high-intensity sports like volleyball. However, the results of our study, while demonstrating improvements from using the ball, contradict the principle of specificity. This principle asserts that training should closely replicate the specific movements of the target activity to optimize effectiveness [36,37]. It is important to note that the greater improvement observed in the group using the ball might be due to various factors, such as the motivational stimulus provided by the introduction of an external element and the increased attentional focus during the exercises. According to some studies, focused attention can stimulate the optimal recruitment of muscle fibers, leading to significant improvements in performance [38,39,40]. Therefore, adopting training approaches that incorporate the use of a ball not only optimizes physical training but may also be effective in enhancing the cognitive and motor skills required in volleyball [41]. Furthermore, the jump test used in our study was non-specific because it was conducted without the ball, as there are currently no validated tests that incorporate its use. This implies that we cannot confidently infer that training with the ball is inherently superior to other types of training. The CMJ may have been particularly suitable because the flexion and extension movements of the lower limbs coincide with the anterior–posterior swing of the upper limbs, promoting the necessary coordination. This coordination is essential for optimizing the speed of jump execution, and consequently, training with the ball may have led to an increase in the jump differential, preparing athletes for game situations that require high performance.

Future studies could benefit from integrating specific tests and activities aimed at investigating this attentional aspect further. While this study contributes insights by examining the impact of a ball in plyometric training, further exploration is warranted to determine if variations lead to more substantial performance gains over longer durations or with larger, more diverse samples. Contrasting these findings with studies that investigate different training modalities or athlete populations could provide a more comprehensive understanding of optimal plyometric protocols for volleyball players. Further studies should also include the analysis of athletes’ perceptions and awareness to measure the effect on jump height improvement. This approach can provide insights that identify the motivation associated with the use of the ball. Understanding how athletes perceive their training will reveal the underlying factors influencing performance and engagement, ultimately leading to more effective training strategies tailored to their needs [42,43].

However, it is essential to acknowledge our study’s limitations. The sample size of 20 participants may have constrained the study’s statistical power, particularly in detecting smaller yet potentially meaningful differences between experimental and control groups. With larger cohorts, future studies could employ more sophisticated statistical analyses to explore nuanced differences in performance outcomes, such as subgroup analyses based on player position or initial fitness levels. Moreover, while the study primarily focused on improvements in jumping ability, future research should broaden its scope to include a comprehensive range of performance metrics. Exploring variables such as agility, reaction time, and specific volleyball skills like blocking or spiking would offer a holistic evaluation of plyometric training’s impact on overall athletic performance and skill proficiency [44,45,46]. Furthermore, extending the intervention beyond six weeks to twelve weeks could provide insights into the long-term sustainability of observed improvements and the persistence of training adaptations. This longitudinal approach would allow for a more in-depth examination of whether initial differences between training protocols become more pronounced over extended periods. Additionally, incorporating biomechanical analyses could elucidate the underlying mechanisms driving performance gains observed in plyometric training studies, further optimizing training strategies for volleyball players.

## 5. Conclusions

Although both groups showed significant improvements in vertical jump performance, the type of plyometric training (with ball vs. without ball) affected the differences in enhancements between the two groups, particularly in the countermovement jump with arm swing. These results could be attributed to the motivational stimulus provided by the ball during training, which positively influenced the athletes’ performance. However, due to the specificity of the sample and the protocols used, further studies are needed to confirm these findings and evaluate their applicability to a larger sample of volleyball players.

## Figures and Tables

**Table 1 jfmk-09-00126-t001:** Training protocol performed by the two groups.

Experimental Group	Control Group
**Frequency:** Three weekly sessions for six weeks	
**Session duration**: Approximately 60–75 min	
**Dynamic warm-up:** Start with a 15-min dynamic warm-up, including exercises such as high knees, and dynamic stretching	
**Core stability exercises:** Includes planks, Russian twists, and leg lifts to m improve players’ overall stability	
– **Initial phase** (first 3 weeks): Three sets of 8 repetitions of jumps with ball block technique, with a 60 s recovery between sets – **Subsequent phase** (last 3 weeks): Four sets of 8 repetitions of jumping with ball block technique, with a 60-s recovery between sets	Performing the same plyometric exercises, but without the use of the ball

**Table 2 jfmk-09-00126-t002:** Anthropometric data of participants.

Variables	Sample (N = 20)
Age (years)	18.6 ± 0.3
Height (cm)	189.8 ± 2.2
Weight (kg)	79.4 ± 1.6
BMI (kg/m^2^)	22.0 ± 1.2

**Table 3 jfmk-09-00126-t003:** Initial performance metrics comparison.

Variable	Experimental Group (Mean ± SD)	Control Group (Mean ± SD)	*p*-Value
SJ (cm)	37.5 ± 1.3	36.9 ± 1.5	0.265
CMJas (cm)	49.2 ± 1.5	48.7 ± 1.6	0.506

**Table 4 jfmk-09-00126-t004:** Comparison of vertical jump performance at the end of the two training programs.

	Experimental Group (*n* = 10)		Control Group (*n* = 10)		Moment × Group Interaction	
Variable	*p*	d (95% CIs)	*p*	d (95% CIs)	*p*	η_p_^2^ (95% CIs)
SJ (cm)	0.001	0.98 (0.72; 0.99)	0.001	0.99 (0.94; 0.99)	0.177	0.09 (0; 0.76)
CMJas (cm)	0.001	0.98 (0.76; 0.99)	0.001	0.97 (0.63; 0.99)	0.004	0.37 (0; 0.91)

## Data Availability

All data generated or analyzed during this study have been included within the manuscript.

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
