# Peer review of "The Effects of the Use of Plyometric Exercises with and without the Ball in the Development of Explosive Strength in Volleyball"

_jfmk, 2024, doi:10.3390/jfmk9030126_

Round 1

Reviewer 1 Report

Comments and Suggestions for Authors

Basic reporting

Dear authors, the manuscript is generally well-written and easy to read; a slight spell-check is required. I have just some concerns that the authors must address. 

Abstract

Line 22-24: I think it is an overstatement, considering the results obtained and the finding of no significant difference between the two protocols. Furthermore, you should state in the abstract that you didn’t find any significant differences between the two groups.

The results and conclusions, thus organized, may appear misleading by leading one to believe that the plyometric protocol with ball is superior to the one without, when the results show no significant difference between the two protocols.

Introduction

The literature on the subject is sufficiently well summarised. However, it could be useful to add some information about:

-          You mention a gap in the literature regarding the differential effects of incorporating the ball into plyometric training protocols but does not provide specific examples or more detailed analysis of the previous studies that have addressed general plyometric training.

-          You assume that incorporating the ball into plyometric training might confer additional benefits but does not provide a strong rationale or preliminary evidence for this assumption

-          Moreover, what’s you r hypothesis? What you’re expect to observe after your intervention?

-          how the findings of this study could be used to inform public health policies and interventions aimed at improving child health outcomes in Brazil and Mozambique.

Methods

Here some personal concers:

-          The sample size is small have you performed an a priori statistical power analysis?

-          How did you have randomized the subjects?

-          A table with the specific protocol followed could be useful.

Validity of the findings

The association between explosive strength and the potential reduction in injury risk, in this case, is speculative as it is an unmeasured variable. I think it should not be indicated as one of the study's objectives.

Line 212: participants are 30 or 20?

A single conclusion section could be useful.

Comments on the Quality of English Language

Minor editing of English language required

Reviewer 2 Report

Comments and Suggestions for Authors

Thank you for inviting me to review this manuscript for your journal. The study by Altavilla et al. addresses a relevant topic for volleyball training and performance, namely the development of vertical jump capacity. However, the work presents some aspects that need clarification:

Why do the authors include injury prevention as an objective of the study if this variable has not been measured?

The finding that training with a ball improves results in SJ and CMJ is counterintuitive and, in this reviewer’s opinion, lacks a solid scientific (and rational) basis. Although introducing an external focus may be more motivating, it clearly creates a distraction for the development of jumping capacity (i.e., the timing of the jump must be adapted to the ball, requiring greater motor coordination to respond to this stimulus). Were the evaluators blinded?

The statistical analysis is insufficient. Why did the authors not include a repeated measures ANOVA to study the interaction between the groups over time?

The results presented in lines 168-175 are incorrect and should be removed until the statistical analysis is properly conducted.

The study generally lacks references to previous studies on the topic and does not compare the results against any similar studies. Common sense suggests that if you test on a non-specific test (SJ or CMJ), non-specific training should yield better results, whereas if you evaluate in a real-world setting (during play, for example), training with a ball might show better results due to the ecological nature of the training and improvements not only in jumping capacity but also in the coordination of the specific movement and adaptation to the external focus (the ball).

The results of this study raise doubts, and the statistical analysis needs to be rewritten. Additionally, the authors' interpretation of their results in the discussion should be modified when the ANOVA results are included.

Comments on the Quality of English Language

It is OK

Round 2

Reviewer 1 Report

Comments and Suggestions for Authors

The authors adressed all my concerns. I have no fhurter suggestions.

Author Response

Thank you for your thoughtful feedback and for taking the time to review our manuscript. We’re glad to hear that we addressed all your concerns and appreciate your support.

Reviewer 2 Report

Comments and Suggestions for Authors

I appreciate the effort the authors have put into addressing the raised issues and improving their manuscript. I would like the authors to include a brief paragraph that qualifies the significance of their main finding. It is crucial to discuss, in an unbiased manner, that the greater improvement observed in this study for the group that trained with plyometrics and a ball contradicts the principle of specificity. Therefore, this result could be attributed to other factors, such as the motivational stimulus provided by an external element during training. Additionally, it would be advisable for the authors to avoid misinterpreting the principle of specificity to justify their finding, given that the jump test employed is non-specific (i.e., without a ball). The other hypothesized factors, such as improvements in neuromuscular coordination or motor control, also do not sound convincing.
It is crucial that the authors soften the possible generalization of the finding of their study, especially in the abstract, as many readers, like me, might still consider such a result to be counterintuitive.
